# Intrinsic Fluorescence Markers for Food Characteristics, Shelf Life, and Safety Estimation: Advanced Analytical Approach

**DOI:** 10.3390/foods12163023

**Published:** 2023-08-11

**Authors:** Ksenija Radotić, Mira Stanković, Dragana Bartolić, Maja Natić

**Affiliations:** 1Institute for Multidisciplinary Research, University of Belgrade, Kneza Višeslava 1, 11000 Belgrade, Serbia; mira.mutavdzic@imsi.rs (M.S.); dragana.bartolic@imsi.rs (D.B.); 2Center for Green Technologies, University of Belgrade, Kneza Višeslava 1, 11000 Belgrade, Serbia; 3Faculty of Chemistry, University of Belgrade, Studentski trg 12-16, 11000 Belgrade, Serbia; mmadnic@chem.bg.ac.rs

**Keywords:** food analysis, intrinsic fluorophores, fluorescence spectroscopy, excitation–emission matrices, chemometrics

## Abstract

Food is a complex matrix of proteins, fats, minerals, vitamins, and other components. Various analytical methods are currently used for food testing. However, most of the used methods require sample preprocessing and expensive chemicals. New analytical methods are needed for quick and economic measurement of food quality and safety. Fluorescence spectroscopy is a simple and quick method to measure food quality, without sample preprocessing. This technique has been developed for food samples due to the application of a front-face measuring setup. Fluorescent compounds–fluorophores in the food samples are highly sensitive to their environment. Information about molecular structure and changes in food samples is obtained by the measurement of excitation–emission matrices of the endogenous fluorophores and by applying multivariate chemometric tools. Synchronous fluorescence spectroscopy is an advantageous screening mode used in food analysis. The fluorescent markers in food are amino acids tryptophan and tyrosine; the structural proteins collagen and elastin; the enzymes and co-enzymes NADH and FAD; vitamins; lipids; porphyrins; and mycotoxins in certain food types. The review provides information on the principles of the fluorescence measurements of food samples and the advantages of this method over the others. An analysis of the fluorescence spectroscopy applications in screening the various food types is provided.

## 1. Fluorescence Spectroscopy in Relation to Food Analysis

The structural characteristics of food are an important quality parameter. For example, texture is crucial for evaluating the quality of cheeses. It depends on the cheese structure at the molecular level [1]. Concerning food structure, an essential quality feature is the health and safety of the food. Government agencies and food industries tend to produce healthy and safe food, by quality control and quality assurance [2,3]. Food is a complex matrix of proteins, fats, minerals, vitamins, and other components, including water. Thus, various analytical methods are currently used for food testing. However, most of the used methods for obtaining these parameters require sample preprocessing and expensive chemicals for measurement. New analytical methods are needed for the quick and economic measurement of food quality and safety [4]. Fluorescence spectroscopy is a simple and quick method to measure food quality, without sample preprocessing.

The analysis of food properties and their changes has been facilitated by the development of the new spectroscopic methods and chemometric tools. Thus, fluorescence spectroscopy has proven to be a promising analytical method for screening various food products [5]. It is a sensitive spectroscopic method since fluorescent compounds–fluorophores are highly sensitive to their environment. Each fluorophore is characterized by an excitation and emission spectrum [6], which may indicate changes in the composition or structure of the specific molecule. This technique has been developed for food samples due to the application of the front-face measuring setup in fluorescence measurements, which allows spectra recording on the surface of turbid or solid intact samples and avoids inner-filter effects caused by spectral distortions [6,7]. The fluorescence steady state emission spectrum of a complex food sample may be a sum of two or more individual components corresponding to various fluorophores. Determining the number and emission profiles of components in an integral spectrum is a prerequisite for obtaining a specific insight into their structure and complementary to the other analytical techniques. This can be achieved by the measurement of a series of emission spectra at different excitation wavelengths in a wavelength range, thus obtaining excitation–emission matrices (EEMs) that are subsequently analyzed by using advanced statistical methods [8]. Such an approach provides information about molecular structure and its changes. Measurement of EEMs for tissue samples in vitro, combined with multivariate chemometric tools, such as parallel factor analysis (PARAFAC) and partial least squares (PLS) regression, is a fast and noninvasive technique relying on endogenous fluorophores to obtain information on food structure and its changes [9]. Synchronous fluorescence spectroscopy (SFS) is a very advantageous screening mode used in food analysis, combining fluorescence EEM spectroscopy and classical fluorescence single spectroscopy [10]. Figure 1 illustrates the measurement of a food sample using spectrofluorometric measurement. 

## 2. Fluorescence Markers in Food

Many nutrients contained in food are naturally fluorescent, e.g., vitamins, amino-acids, and polyphenols, as well as newly formed products developed in Maillard reaction and lipid peroxidation. Table 1 shows intrinsic fluorescent markers in food and their excitation and emission maxima.

The endogenous fluorophores in food are the amino acids tryptophan and tyrosine; the structural proteins collagen and elastin; the enzymes and co-enzymes NADH and FAD; vitamins; lipids; and porphyrins. Certain food types contain mycotoxins as products of pathogenic fungi. The excitation maxima of the food fluorophores are in the range of 250–450 nm, while the emission maxima are in the range of 280–700 nm [11]. At excitation wavelengths of 275–295 nm, tyrosine and tryptophan are fluorescent with emission maxima at 320–350 nm [12]. Collagen, a fibrous protein, is the major extracellular matrix component that is present in nearly all organs. Collagen fluorescence originates from cross-links among the three composite fibrils and the pyridoline ring. Collagen fluorescence has an excitation maximum at 325 nm and an emission maximum at 400–480 nm, depending on the number of cross-links [44]. The emission of tryptophan is highly sensitive to its local environment and is thus often used as a reference group for protein structure changes, the binding of ligands, and protein–protein associations. The emission fluorescence region (305–450 nm) allows for the study of the fluorescent Maillard-reaction products (maximum emission at 440 nm) [57]. Riboflavin has a strong and broad fluorescence emission peak in the region of 525–531 nm (excitation wavelength 380 nm). In ultraviolet light, riboflavin is degraded into two fluorescent products: lumichrome and lumiflavin, with emission maxima between 444–479 nm and 516–522 nm, respectively. The reduction in the fluorescence at about 525 nm might reflect the photo-degradation of riboflavin [21,22]. The (poly)phenols have an excitation maximum at 360 nm and emission maxima in the range 400–500 nm [23,48]. The excitation and emission of aflatoxins are at 360 nm and 400–460 nm, respectively. As fluorescence emission spectra of particular fluorophores in a food sample can overlap, a combination of measurements of EEMs (Figure 2a,b) and advanced statistical methods is used to obtain pure spectra of particular fluorophores (Figure 2c) and their relative contribution to the integral spectra.

## 3. Applications of Fluorescence Spectroscopy in Food Analysis

### 3.1. Meat, Fish, and Eggs

Fish and meat are fragile food products, and their quality deteriorates during shelf life due to microbial growth, oxidative, and enzymatic processes [24,25]. On the other side, food freezing is a method of long-term storage, but thawing can cause biochemical and physical changes that might affect the characteristics of meat, fish, and other seafood. The establishment of a rapid and reliable method for the distinction between fresh and thawed meat products is essential. Meat, fish, and seafood products are multifluorophoric systems, containing aromatic amino and nucleic acids, nicotinamide adenine dinucleotide (NADH), vitamin A, riboflavin, and oxidation products [10,51].

Fluorescence spectroscopy has been used to detect structure changes in meat, such as collagen contents in adipose and connective tissues of meat, which have significant importance for meat tenderness and textural properties. Variation in myoglobin content hinders the recording of the collagen autofluorescence in the 400–640 nm range after 380 nm excitation. However, collagen content could be determined by applying multivariate regression models on the fluorescent spectral data [45].

When meat is subjected to thermal treatments, many changes occur such as the denaturation of protein, the degradation of some fluorophores (tryptophan residues in proteins and vitamins), and the development of some new fluorophores (Maillard-reaction products, heterocyclic amines), causing changes in the fluorescence signals of meat [13,14,28]). Gatellier et al. (2009) used fluorescence spectroscopy to observe two types of reactions occurring during the heating of meat: the oxidative reaction of proteins with the aldehydes formed by lipid peroxidation and the glycolytic reaction of proteins with sugars with long polyol chains [29]. The two types of products had different fluorescence and were soluble in solvents of different polarities. Simultaneously with the fluorescence study, thiobarbituric acid reactive substances and protein carbonyls were measured. The correlations between these two parameters and the fluorescence emission additionally confirmed that the products of the reaction between proteins and aldehydes produced by lipid peroxidation are the source of fluorescence. The fluorescence results corroborated the increase in meat protein surface hydrophobicity during heating [30]. The results were confirmed in an in vitro study in which aldehydes were added to the amino acids or minced meat; fluorescence of the obtained complexes was assigned to the products of the reaction of proteins with aldehydes [31].

Fluorescence spectroscopy has also been widely used to rapidly and non-destructively detect the microbial spoilage of meat. In a study, Oto et al. (2013) used fluorescence spectroscopy in the EEM mode to evaluate the microbial growth on the pork meat surface, by following changes in Trp and NADH emission [60]. Aït–Kaddour et al. (2011) reported the potential of a portable spectrofluorometer to detect microbial spoilage of minced meat [61]. Fluorescence spectroscopy has been used for testing the lactic acid bacteria spoilage of the sausages [26]. The fluorescence maxima of aromatic amino acids, nucleic acids, NADH, and FAD as intrinsic markers were measured. The chemometric methods PCA and factorial discriminant analysis (FDA) were applied to identify lactic acid bacteria at various levels in the samples.

The oxidative stability of meat products during different processing and storage conditions was evaluated using fluorescence spectroscopy, such as a report of rapid detection of oxidative damage to pork and turkey meat samples [49]. The detection of protein degradation due to oxidation occurs in meat samples. The tryptophan fluorescence of porcine myofibrillar protein was used to obtain information about protein oxidation and modifications in the secondary and tertiary structure of the proteins induced by different thawing methods [32].

Quality indicators (freshness, the presence of various harmful substances) of various white meat products such as fish, shrimp, chicken, duck, and goose may be obtained by fluorescence spectroscopy with the assistance of machine learning and convolutional neural network methods, to prevent the spoilage of meat and cause harm to consumer health and to the ecosystem. Fish is rich in amino acids, vitamins, and minerals (such as phosphorus, calcium, and iron). The quality of fish is mainly dependent on storage conditions. On the other side, antimicrobial compounds are added in aquaculture to inhibit the growth of microorganisms and prevent fish diseases, but their residues may accumulate in fish, causing potential health risks to consumers [27]. Poultry meat should be handled properly to reduce the risk of food poisoning. There are also problems with chicken influenza. Moreover, some harmful substances may be present in poultry meat, such as antibiotics and pesticides, harming human health. Fluorescence spectroscopy was used to detect chloramphenicol (CAP) residues in chicken meat [62]. Another predictive model was developed for the rapid detection of gentamicin residues in duck meat by fluorescence analysis based on the strong fluorescence properties of gentamicin and o-phthal-aldehyde derivatives (OPA) in the presence of emulsifier OP-10 and mercapto-ethanol [63].

The potential of front-face fluorescence spectroscopy for monitoring fish freshness and discrimination between fresh and aged fish was reported on two fatty fish species and two lean fish species, by recording changes in spectral shape in the emission of tryptophan after 290 nm excitation, the emission of aromatic amino and nucleic acids after 250 nm excitation, and NADH emission after 336 nm excitation, as a function of storage time [64]. Changes in the protein structure during processing were investigated by fluorescence spectroscopy [33]. Discrimination between fresh and frozen-thawed fish by fluorescence spectroscopy, through signals of NADH, tryptophan, riboflavin, and vitamin A, was reported [34].

Freshness is a vital factor for consumers of shrimp as it has an important relationship with taste and shelf life [65]. Schneider et al. [66] developed a method for the detection of 10 fluoroquinolones in shrimps, by fluorescence detection and quantification [67]. Fluorescence spectroscopy in EEM mode, combined with multivariate chemometric tools, such as PARAFAC and PLS regression, was applied to evaluate fish oil samples [68].

Principal component analysis and the Mahalanobis distance method applied to the emission spectral data sets recorded on cod, mackerel, salmon, and whiting fillets enabled discrimination between fresh and aged fish fillets. Thus, the fluorescent spectra of intrinsic fluorophores–aromatic amino acids, nucleic acids, and NADH may be considered as fingerprints of fish filet freshness. A device including a fiber optic and a CCD detector would enable fast screening of a large number of samples [64]. Collagen type I and type V make up the major part of the intramuscular connective tissue in fish muscle. The collagen content of fish muscle varies considerably from species to species. The autofluorescence of salmon and cod muscle was found to be similar to that of collagen type I and type V, with the same peaks being obtained around 390, 430, and 480 nm. These similarities are supported by principal component analyses (PCA). Variations of the collagen emission within one fillet were different for different fish species [46]. The application of front-face fluorescence spectroscopy combined with PCA and Factorial Discriminant Analysis (FDA) was studied as a tool for the assessment of table egg freshness during storage at fridge temperature for up to 55 days and at a constant relative humidity. Aromatic amino acids, nucleic acids, fluorescent Maillard reaction products, and vitamin A were considered intrinsic emitters. The best results were obtained with vitamin A fluorescence spectra, showing that vitamin A may be a useful fluorescent intrinsic probe for the assessment of egg freshness during storage [19].

### 3.2. Dairy Products

Certain compounds in dairy products are fluorescent intrinsically or after chemical modification: aromatic amino acids, vitamin A, riboflavin, products of lipid oxidation, nicotinamide adenine dinucleotide (NADH), and Maillard reaction products. These compounds are intrinsic reporters of the quality and shelf life of dairy products under various external conditions. Fluorescence spectroscopy along with the multivariate approach uses combinations of these fluorophores to follow changes in milk products’ properties [52]. Figure 3 illustrates an example of the emission spectra of milk. Under excitations at 290 nm and 330 nm, the emission maxima 350 nm and 410 nm were obtained, corresponding to the proteins’ tryptophan and vitamin A, respectively, based on literary data [8], Table 1. Heat treatment is important in processing dairy products, enabling microbiological safety. However, this treatment may affect the physicochemical properties of the products, such as changes in proteins and heat-sensitive vitamins and the formation of oxidation products (Maillard reaction). Intrinsic fluorescent emitters, such as tryptophan, advanced Maillard reaction products, and riboflavin, may provide information about heat treatment-induced changes. Thus, fluorescence spectroscopy at selected wavelength regions provides information on the level of heat-induced changes in dairy products. Shelf life is another vital property of dairy products that may be checked in a simple way using fluorescence spectroscopy. Despite the long shelf life of the infant milk and milk powders, during storage, their structure may be changed. The intrinsic fluorescence markers may be used for monitoring the chemical and physical properties of dairy products change (protein and lactose degradation) during storage. The presence of adulterant substances in dairy products may be tested by traditional chemical tests, which are time-consuming, but using fluorescence spectroscopy is quicker and does not need sample preprocessing. For example, a fluorescence quenching study was performed to observe the interaction between resveratrol and whey protein products [35]. On the other hand, in cheese production, one of the main phases is the milk coagulation process. The structural changes of proteins and their physico-chemical environment during coagulation may be monitored through tryptophan fluorescence [36]. The physical properties of triglycerides and protein–lipid interactions may be monitored by observing vitamin A fluorescence [53].

The deterioration of the cheese during storage depends on the handling in the post-manufacturing processes. During cheese storage, the changes in structure and composition of the cheese constituents, especially protein and fat, may occur due to changes in external conditions such as light exposure and varying temperature. Front-face fluorescence spectroscopy was applied for an evaluation of the stability of cheese during storage. For cheese samples, EEMs were obtained and analyzed using PARAFAC, for different storage durations and temperatures. Tryptophan and vitamin A were found to be fluorophores related to the storage conditions, showing that fluorescence spectroscopy with chemometrics may provide fast and simultaneous determination of chemical changes of these compounds in cheese [15].

Fluorescence spectroscopy may provide the cheese quality parameters such as rheological properties data. The molecular and macroscopic changes of ripened soft-cheese samples, obtained in different manufacturing processes, were studied from the surface to the center of the cheese. For each cheese, the fluorescence and rheology data were analyzed using PCA, FDA, and CCA. The discrimination of the data was performed by FDA performed on the vitamin A and tryptophan fluorescence spectra. Correlations between the rheological and fluorescence data for proteins (tryptophan) were high and indicated that the molecular (fluorescence) and the macroscopic (rheology) changes in the texture of cheese can be related as a function of their location in the cheese [16,17].

The excitation spectra of vitamin A provide information on the protein-fat globule interactions during milk coagulation during cheese production. The shape of the vitamin A excitation spectrum was related to the physical state of the triglycerides in the fat globule [57]. To follow the changes in the cheese structure during ripening, Trp and vitamin A were monitored as intrinsic fluorophores. The red shift of the tryptophan emission maximum was observed in the ripened cheese (60 days old), due to exposure of Trp residues to the aqueous phase. The excitation spectra of vitamin A changed considerably during cheese ripening, due to the physicochemical changes in the environment and triglycerides in the fat globules [54]. The degradation of riboflavin in cheese during storage can be monitored through the formation of the fluorescent products: lumichrome and lumiflavin. The reduction in the riboflavin emission maximum at about 525 nm may be an indicator of its photo-degradation [21,22].

It was shown that fluorescent spectroscopy can be used to differentiate textures among investigated cheeses due to the different molecular structures of each cheese forming during ripening. Protein tryptophan emission spectra and vitamin A excitation spectra were measured on several soft cheese samples in front-face fluorescence configuration. The multivariate statistical analysis (PCA and FDA) was applied to differentiate between the cheese samples and to observe changes in cheese texture during ripening. The characteristic wavelengths that most influence separation of the spectra were derived from the spectrum associated with the principal components. They are related to the changes in protein structure [37].

Fluorescence spectroscopy in combination with chemometrics was used to test the oxidative stability and quality of yogurt. The front-face fluorescence spectra of yogurt samples were recorded during storage and in two different packaging materials, with the excitation 270 to 550 nm and emission 310 to 590. A correlation was found between the fluorescence spectra of yogurt and riboflavin content determined by the standard AOAC method [50]. The EEMs and PARAFAC were applied to monitor the three fluorophores (tryptophan, riboflavin, and lumichrom) in yogurt as markers for storage conditions. The excitation range 270 to 550 nm and emission range 310–590 nm were used in the front-face measurement configuration. The effect of light treatment and the type of package was studied [69].

### 3.3. Fruits and Seeds

Front-face fluorescent spectroscopy was used for the classification of fruits, using pineapple as a model, based on structural modifications of pulp and skin, generated by physiological disorder caused by plant stress or plant infection. The EEMs were obtained and subjected to the N-CovSel method, a type of discriminant analysis, to discriminate between samples with respect to their fluorescence spectra and class. The most relevant features extracted were those with the emission ranges 250–450 and 600–700 nm, being correlated with amino acids, phenolic compounds, and oxidation products, and with the pigments, respectively [70].

Fluorescence spectroscopy was used to separate and quantify diseases and damage (scab and rot) on the surface tissues of fruits and vegetables, using the examples of apples and potatoes. The fluorescence spectra of surface slices of healthy or damaged (mechanically or infected) apples and potatoes were recorded. The characteristic excitation–emission regions were observed for healthy and damaged apples and potatoes. The spectra of the healthy surface of both the apples and potatoes differed from those measured on the damaged areas. The healthy and diseased samples could be successfully separated by PCA applied on the emission spectra. The results indicate that fluorescence spectroscopy can be used to detect and separate diseased and healthy fruits and vegetables [71].

Front-face total synchronous fluorescence (TSF) in combination with MCR-ALS was used to differentiate differently processed apple juices. Synchronous fluorescence measurements enabled increased selectivity in comparison with the conventional emission measurements. The emission profiles of five fluorescent components with different spectral profiles and contributions to the total fluorescence were observed. Although MCR-ALS in combination with TSF may provide a selective tool for relationships between the fluorescence and the total antioxidant indices of the apple juices, the evaluation of the possibility and power of fluorescence for the prediction of the total antioxidant indexes requires further study [72].

Grape maturity on both red and white grapes was studied on intact grape berries of Cabernet Sauvignon and Merlot using the multiplex fluorescence sensor [73]. It is based on the detection of fluorescence emitted by chlorophyll (Chl) in the red (RF) and far-red (FRF) spectral regions, under excitation with different LED sources in the UV and visible spectral range. The intensity of the chlorophyll fluorescence (ChlF) is measured. The light emitted from Chl in a grape berry depends on the amount of excitation light able to reach the Chl in the chloroplasts. Anthocyanins, localized in the layers above the ChL containing layers, can weaken part of the incident light before it reaches the Chl molecules. Therefore, the higher the anthocyanins concentration, the lower the ChlF intensity. The extent of the anthocyanins attenuation also depends on the wavelength of the excitation beam. Anth absorbs mainly around 520 nm, and the green spectral range will be attenuated more than red light, in which range anthocyanins’ absorption is low. Thus, the measured ChlF is significantly lower at green excitation than at red excitation. A comparison of the two fluorescence signals gives an index proportional to the berry skin anthocyanins content [74]. As a step forward, the absorbance-transmission and fluorescence excitation–emission matrix (A-TEEM) spectroscopy method has been developed, which simultaneously records the individual excitation and emission spectra for all fluorescent sample components and gives information on non-fluorescent components that absorbed light. This method also provides information necessary to correct the fluorescence spectra for inner-filter effects, thus providing quantitative fluorescence spectral information that is mainly independent of component concentration [75]. This method was used to predict maturity indices on the Cabernet Sauvignon grapes produced under four viticulture treatments during two growing seasons [76]. The fused A-TEEM data were combined with a machine learning technique to predict important grape maturity indicators (3-isobutyl-2-methoxypyrazine, pH, total tannins, total soluble solids, and malic and tartaric acids) during the grape ripening period. The classification of grape maturity was performed using extreme gradient boosting XGB regression discriminant analysis. The A-TEEM measurements were performed at the 240–700 nm excitation and 242–824 nm emission alongside absorbance measurements at 240–700 nm. The 520 nm absorbance, related to the compounds contributing to the color of red wine grapes, was used as the red color classification variable. Different fluorophores have been considered intrinsic markers in native olive oil and recorded by front-face fluorescence spectroscopy. The spectra provided information about the type and amount of emitters, mainly (poly)phenols, vitamins, riboflavin, and chlorophyll [77]. Fluorescence spectroscopy combined with (parallel factor) PARAFAC analysis was tested to monitor the changes in virgin olive oil during storage under moderate conditions from a multidimensional perspective. The spectra of oil samples were measured using the right-angle geometry. The oil samples were diluted in n-hexane (3% *v*/*v*) to avoid spectral distortions. The inner filter effect was corrected based on the simultaneous absorbance measurements. The EEMs were recorded and for all sample concentrations of fluorophores (phenols, tocopherols, and chlorophyll pigments) and of physical–chemical parameters (peroxide value, free acidity), and sensory attributes were measured. The six components were extracted by PARAFAC, two components related to chlorophyll pigments; the components assigned to tocopherols, phenols, and oxidation products were selected for their suitability to discriminate between fresh and aged oils [78]. Virgin olive oil (VOO) is derived from the olive fruit by mechanical maceration without any chemical procedure. Thus, olive oil is an expensive product and is often the subject of adulteration with cheaper vegetable oils. Adulteration is mostly detected using HPLC and GC-MS, by measuring tocopherol content and the analysis of fat acids. The fluorescence spectra of vegetable oils are noticeably different from the olive oil spectra. Fluorescence spectroscopy was used to detect the adulteration of 12 types of olive oil samples with vegetable oils by recording the emission spectra in the 350–720 nm range for 375–450 nm excitations, using an optical fiber. Fatty acid and tocopherol contents were analyzed in parallel. The samples differed in the emission spectra shape, as well as the number and positions of the emission maxima. Based on the spectral and chemical analysis, certain samples were found to be mixtures of sunflower and olive oils, while some of them were natural olive oils. Thus, fluorescence spectroscopy has been shown to be suitable for the rapid detection of olive oil adulterations [79].

Fluorescence and synchronous fluorescence spectroscopy (SFS) techniques were used to characterize and distinguish various edible oils. The oil samples were dissolved in n-hexane. The fluorescence emission peaks were assigned to tocopherols and chlorophylls. There was a significant difference among the oil samples in the SFS spectra. The K-nearest-neighbor (K-NN) and LDA methods were performed to compare the sets of SFS spectra of different oils. Successful discrimination between the oil classes was obtained [80].

Fluorescence spectroscopy (EEMs) has been applied for screening the antioxidative capacities of soybean seeds by recording spectral changes in the UV-irradiated soybean (Glycine max L.) seeds [81]. The evaluation of soybean protein and oil content in different soybean flours was performed using the EEMs measured in front-face configuration. Also, the precision of the protein and oil content prediction was investigated. The second derivative synchronous fluorescence (SDSF) spectra were extracted from the EEMs. The partial least square regression and support vector machine models were developed on each of the EEMs and SDSF spectra. Based on the loading spectra, fluorescence excitation/emission combinations that contribute most to the prediction of protein and oil content were determined [38].

Front-surface fluorescence spectra of flours from different cereals (wheat, rice, and maize) were recorded. The three emission bands were observed in the 290–600 nm spectral range, after the 280, 330 nm, and 450 nm excitation, and were assigned to the cereal proteins, tocopherols/pyridoxine, and xanthophylls, respectively. The fluorescence spectra of flours depended on the species of their origin [39]. In another study, PARAFAC was applied on the front-face measured EEMs of the flours of various cereals. The two spectral regions were recorded, similar to the Zandomeneghi’s study. Similarly, proteins, tocopherols, pyridoxine, and 4-aminobenzoic acid were identified as the intrinsic emitters. PARAFAC was used to construct the partial least squares discriminant analysis (PLS -DA) classification model using scores to quantify the differences between the flour types. Rice and corn showed the lowest cross-validated classification error, while spelt and wheat had somewhat higher values. This result indicated that these four flour types contained specific levels of four modeled components. The flour types that contained small number of samples (barley, buckwheat, oat, and rye) showed lower classification success rates [56]. As a step forward, a front-face synchronous fluorescence spectroscopy (FFSFS) study was performed for the composition investigation of binary and quaternary blends of maize, sorghum, wheat, and barley flour. Prediction models were created by partial least square (PLS) regression, providing plausible results. The study showed that fluorescence spectroscopy combined with suitable chemometric tool may be used for the reliable composition analysis of multiple cereal flour blends, avoiding the time-consuming procedure of sample preparation [82]. Aflatoxins, produced by the fungi Aspergillus flavus and Aspergillus parasiticus, contaminate various food products, such as seeds, dried fruits, and nuts. These products can be infested with toxigenic fungi in the field or during storage and thus be contaminated with mycotoxins, exhibiting toxic effects in animals and humans. The fluorescence EEMs coupled with the MCR-ALS technique is shown to be an effective and rapid method for the estimation of the degree of aflatoxin B1 (AFB1) contamination in maize flour or seeds. The MCR-ALS analysis revealed two components in the emission spectra (400 nm–445 nm range) of all of the contaminated and uncontaminated flour samples, characteristic of phenolic compounds. In the AFB1 contaminated samples of the maize flour fluorescence emission spectrum, the shape changed compared with the uncontaminated samples. The components’ positions were red-shifted in the AFB1 contaminated flour samples compared to the uncontaminated ones. It was found that the ratio of the areas of the two spectral components is proportional to the intensity of contamination and thus could be an indicator of the AFB1 contamination degree [20]. In a related fluorescent study, optical fiber fluorescence spectroscopy and multispectral imaging (MSI) were used for the discrimination of maize seeds naturally contaminated with AFB1 and uncontaminated maize seeds. The differences in the spectral shape and the position of emission maximum between the AFB1-contaminanted seeds and the control were notable (Figure 4).

Using linear discrimination analysis to analyze fluorescence data, as well as normalized canonical discriminant analysis (nCDA) applied to the MSI data, uncontaminated and AFB1-contaminated seeds were discriminated with 100% classification accuracy [59].

### 3.4. Honey

Honey contains two minor but important components: proteins [40] and phenolic compounds [83], which are intrinsic fluorophores. Phenolics in honey originate from nectar, while proteins mainly originate from bees (2/3th of the total honey proteins) and pollen suspended in nectar (1/3th of the total honey proteins [40]. Fluorometric ratiometric analysis may be used for the fast and reliable screening of honey sample variability and the selection of the samples for further, more detailed analysis. The front-face spectroscopy was used to obtain the EEMs of the honey samples obtained from different beekeepers, and subjected to MCR-ALS method. The two characteristic emission ranges were observed, 340–360 nm and 415–450 nm, assigned to the proteins and phenolics in honey, respectively. The area of the characteristic protein (Pr) and phenolic (Ph) components in the emission spectra of lime honey were calculated as a ratiometric indicator of the honey samples foraged from various beekeepers. The fluorescence method was validated by other analytical methods. The different values of the (Ph/Pr) ratio after extraction and after packaging, as well as the absence of protein component after MCR analysis or poor expression of the protein maximum in the raw spectra, may be connected to the variability in the properties of honey obtained from different bee colonies, or to the inappropriate homogenization of honey before it was packed in jars [41]. In another study, the ratio of the honey emission spectral components linked to the proteins and phenolics in honey was reported to be a ratiometric indicator of variability in honey samples collected in different seasons. It was also found, based on the spectral analysis, that oxalic acid, a protective agent, does not influence the proteins’ and phenolics’ composition of honey, indicating that is not harmful to the bees [42]. Front-face fluorescence spectroscopy was reported to be a suitable tool for the classification of honey samples according to their botanical origins. The PCA combined with factorial discriminant analysis (FDA) was used for the processing of fluorescence data obtained after excitation set at 250 nm (emission: 280–480 nm), 290 nm (emission: 305–500 nm), and 373 nm (emission: 380–600 nm) and emission set at 450 nm (excitation: 290–440 nm). The spectra were assigned to the proteins, phenols, and Maillard reaction product, respectively. The FDA was applied to the PCA on each data table and led to discrimination between the investigated honey types [43]. In another study performed by front-face fluorescence spectroscopy combined with LDA, differences in geographical origin were studied within the groups of honey samples of the same botanical origin, which were collected in different countries. A statistically significant difference was observed by using MANOVA among all honey types of different geographical origins. The samples were successfully classified by LDA according to their geographical origin, showing that front-face fluorescence spectroscopy is a suitable method for the assessment of the geographical origin of a particular unifloral honey [84]. In another study, fluorescence spectroscopy combined with advanced statistical methods, parallel factor analysis PARAFAC, and PLS-DA, was used for honey characterization and classification. The excitation and emission ranges were 240–500 nm and 270–640 nm, respectively. The PARAFAC model was used to obtain the number of fluorophores in honey and their relative concentration. The emission peaks were assigned to the phenolic compounds and Maillard reaction products, which exhibited great differences among the types of honey of different botanical origins. The detection of honey adulteration, based on the PLS-DA classification model, was with 100% sensitivity and specificity [58].

The MCR-ALS method was used to analyze the EEMs of multifloral honey samples, to estimate the extent of the infection of the corresponding honey bee colonies with *Nosema ceranae* or *Varroa destructor*. The ratio of the spectral components originating from proteins and phenolics in honey was a ratiometric indicator of the infection level in related hives [18]. For honey samples originating from the colonies infected with *V. destructor*, the parameters of fluorescence emission spectra were related to the protein and phenolic content in honey and to the enzymes that originate either from the plant source (catalase) or from the bees (diastase). The EEMs of the honey samples were decomposed using PARAFAC. Figure 5 shows the emission and excitation features of the loading vectors of the three PARAFAC components in the form of the EEM heatmaps. The characteristic regions correspond to the phenolic compounds (Figure 5a,b), and aromatic amino acids/proteins (Figure 5c). The infestation level was highly positively correlated with the catalase activity and the PARAFAC spectral component linked to the phenolics in honey [85]. These results indicate that PARAFAC phenolic component, together with the catalase activity, may be a promising marker for *Varroa destructor* infestation levels of colonies, by screening corresponding honey samples.

### 3.5. Biscuits

Front-face fluorescent spectroscopy combined with chemometric methods was applied to evaluate the quality of biscuits during storage. Vitamin A and Trp spectra, after PCA processing, provided discrimination between the samples according to the type of antioxidants used in production, while the spectra of fluorescent Maillard reaction products spectra allowed for discrimination between the samples regarding the storage time. Differentiation between the samples regarding both used antioxidants and shelf life was obtained using joint common components and specific weights analysis (CCSWA) applied to the fluorescence, physicochemical, textural, and colorimetric measurements. The partial least-squares regression (PLSR) applied to the fluorescence data gave a good prediction of water activity, hardness, and moisture content [55].

## 4. Future Directions and Perspectives

Fluorescence spectroscopy has proven to be a promising method to study changes in food properties during shelflife and the detection of adulteration. However, additional research is needed for the broader application of this method for safety and quality testing in food processing. Regarding the use of intrinsic fluorophores to monitor the effects of environmental factors on the food structural changes, additional studies are required to validate the findings, as well as understand the effect of various factors including pH and temperature. The results obtained by fluorescence/chemometrics methods remain to be validated with chemical and physicochemical reference analyses to prove the identification of the fluorophores and their use in monitoring changes in tested food samples. A combination of hyperspectral imaging with fluorescence spectroscopy will be a promising advancement in spectroscopic methodology dedicated to food sample screening [86]. The application of chemometrics methods in processing fluorescence data plays a vital role in the further development of fluorescence-based monitoring of food samples. The introduction of new methods such as artificial neural network technology in fluorescence spectral data processing is promising. This was recently documented in the detection of pesticide residues of acetamiprid on solid surfaces based on fluorescence spectral data [87]. Also, the use of artificial intelligence, such as machine learning, in chemometric techniques may considerably accelerate the analysis of big data collected using fluorescence methods. Another step forward will be enabled by the construction of portable devices for the fluorescence measurement of food samples in the field, which has been under development for different kinds of samples (meat, milk, and seeds). Such types of devices have already been constructed for applications on vegetables and fruits, such as Dualex Multiplex, and FluorPen for the detection and quantification of fluorescent and non-fluorescent compounds such as phenolic compounds, flavonoids, anthocyanins, and chlorophyll, in different types of samples [47,88,89].

Fluorescence spectroscopy has some limitations; one of them is its constraint to the fluorescent compounds. Nonfluorescent matrix components may be affecting the emitters in the sample, but this influence may also be used to extract some information on the present nonfluorescent compounds. Another limitation may be a variation in the emission properties of the food components highly fluorescent components, possibly suppressing emission from components with low-emitting compounds. Although the fluorescence measurements are generally fast and simple, the EEM acquisition on a conventional spectrofluorometer may have a certain duration. The new instrumentation might reduce the acquisition times. The selection of the most suitable measurement technique and corresponding chemometric method is also important. For example, front-face measurements are suitable for samples with high optical density. Unfortunately, such measurements are limited to the sample surface and cannot measure the bulk of the sample. Another important issue is obtaining the truly corrected fluorescence spectra since we need to compare spectra measured on different instruments. Considering all of these issues, a careful approach is needed in practical applications [90].

## Figures and Tables

**Figure 1 foods-12-03023-f001:**
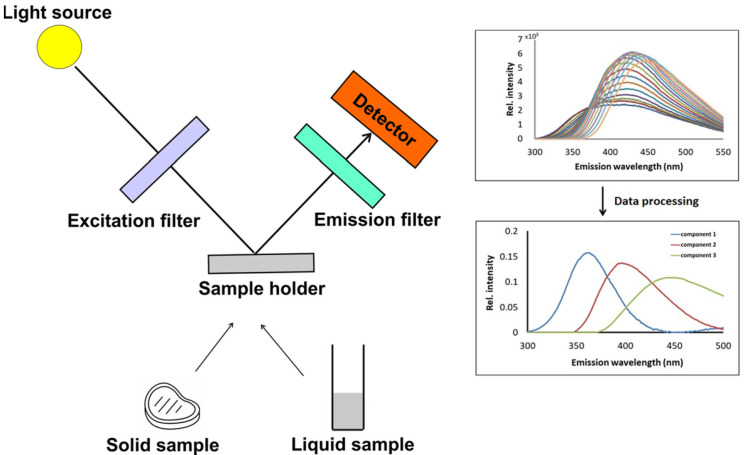
Schematic presentation of the setup in the front-face fluorescence measurement of a solid or liquid food sample. The obtained spectral data (excitation–emission matrix) and their corresponding chemometric processing are also shown.

**Figure 2 foods-12-03023-f002:**
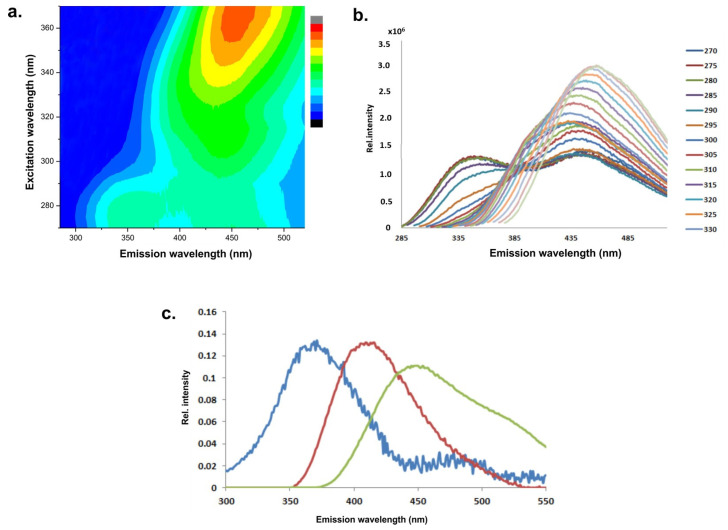
Schematic presentation of the pathway to obtain pure spectra of the food-containing intrinsic emitters. An example of a honey sample: (**a**,**b**) EEM in 2D and 1D presentation, (**c**) estimated emission profiles obtained by MCR-ALS.

**Figure 3 foods-12-03023-f003:**
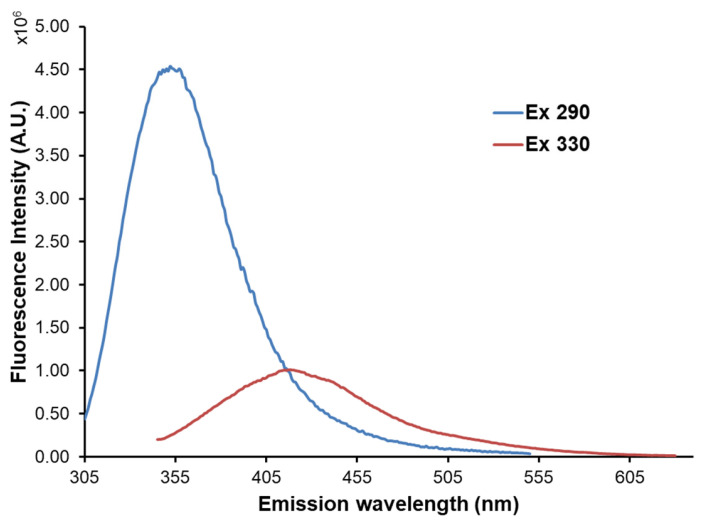
Emission spectrum of the non-fat milk. The emission maxima, under excitation at 290 nm and 330 nm, correspond to tryptophan (blue) and vitamin A (red), respectively, as intrinsic markers.

**Figure 4 foods-12-03023-f004:**
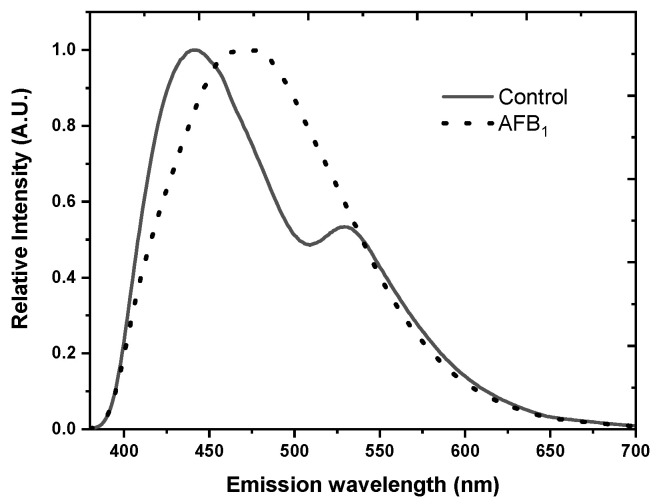
The normalized fluorescence emission spectra obtained on the whole maize seed using the optical fiber. Solid and dashed lines denote control and aflatoxin B1-contaminanted Zea mays seeds, respectively. Excitation wavelength: 340 nm (reproduced from reference [59]).

**Figure 5 foods-12-03023-f005:**
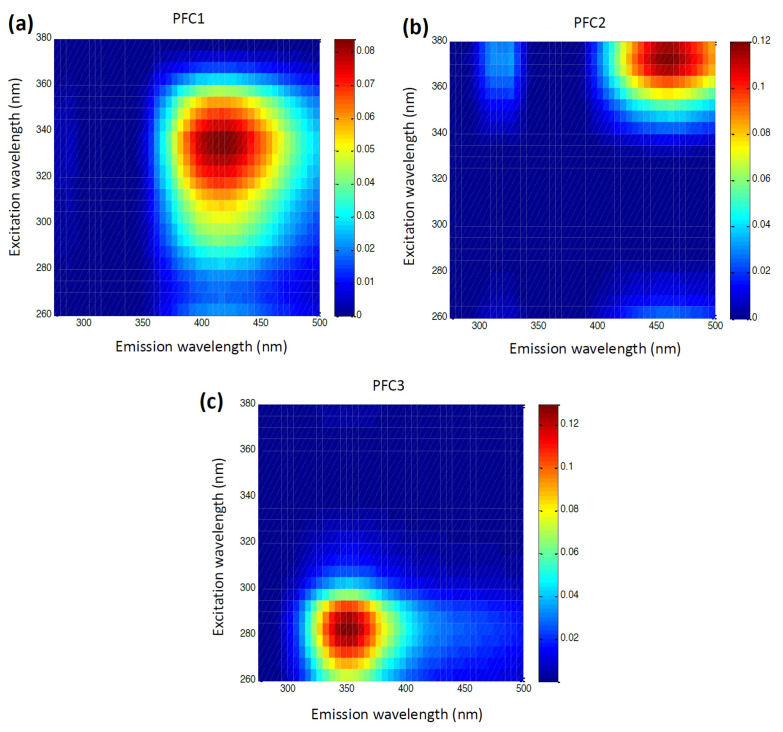
EEM heatmaps for the first (**a**) and the second (**b**) PARAFAC component corresponding to phenolics, and the third PARAFAC component (**c**) corresponding to proteins. Intensity scale is in arbitrary units (reproduced from reference [85]).

**Table 1 foods-12-03023-t001:** Intrinsic fluorescent markers in food.

Intrinsic Fluorophores	Excitation Maxima (nm)	Emission Maxima (nm)	References
*Aminoacids*			
Tryptophan	280	350	[6,10,11,12,13,14,15,16,17,18,19,20,21,22,23,24,25,26,27]
Tyrosine	275	305	[11,12]
Phenylalanine	260	280	[11,12]
*Proteins*, *enzymes*, *coenzymes*			
Proteins	260–280	320–350	[11,12,13,14,15,16,17,18,28,29,30,31,32,33,34,35,36,37,38,39,40,41,42,43]
Collagen	325	400–470	[11,44,45,46]
Elastin	325	400–425	[11]
Flavins	380–450	515–535	[19,27,28,29,34,35,47,48,49,50]
NADH, NADPH	330–350	440–465	[10,11,19,34,35,40,47]
*Vitamins*			
Vitamin A	320–330	470–510	[10,15,16,17,19,34,37,51,52,53,54,55]
Vitamin B6 forms	315–340	385–425	[11,39,56]
Vitamin B12	250–275	305–335	[11]
Vitamin C	310–360	400–440	[11]
Vitamin D	390–425	450–480	[11]
Vitamin K	245	430–440	[11]
*Lipids*			
Lipofuscin	340–400	350–600	[11]
Maillard reaction products	350	440	[13,14,19,28,35,43,52,55,57,58]
*Carotenoids*	450–480	525–580	[47]
*Phenolics compounds*			
Ferulic acid	260	420	[48]
Caffeic acid	260	425	[48]
Catechin, epicatechin, and vanilic acid	278	360	[41]
*trans*-Resveratrol	330	375	[35,41]
*Aflatoxins*	360	400–460	[20,59]

## Data Availability

No new data were created or analyzed in this study. Data sharing is not applicable to this article. The data presented in this study are available on request from the corresponding author.

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
