# Peer review of "Intrinsic Fluorescence Markers for Food Characteristics, Shelf Life, and Safety Estimation: Advanced Analytical Approach"

_foods, 2023, doi:10.3390/foods12163023_

Round 1

Reviewer 1 Report

In this review paper, the authors summarized the advances in intrinsic fluorescence markers for food testing including characteristics, shelf life and safety estimation. The manuscript was rationally-organized but not comprehensively described. Current version is too concise, many references had been simply introduced by only one or two sentences (for examples: ref: 23, 53-55). It is suggested to extend the content by providing more figures and mechanism depiction to better present the listed examples. In addition, we don’t think that Figure 1 is suitable for the proposed topic. Overall, it needs to address aforementioned issues carefully.

Moderate editing of English language required

Author Response

Reviewer 1

In this review paper, the authors summarized the advances in intrinsic fluorescence markers for food testing including characteristics, shelf life and safety estimation. The manuscript was rationally-organized but not comprehensively described. Current version is too concise, many references had been simply introduced by only one or two sentences (for examples: ref: 23, 53-55).

Answer: In the revised version, we included more comprehensive explanations of the existing references, such as 23, 53-55, but also for some other references. Also, we provided additional examples with extensive explanations . The new text is added as track changes.

It is suggested to extend the content by providing more figures and mechanism depiction to better present the listed examples.

Answer: We added the new Figures (Figs 3 and 4) along with the corresponding explanations.

In addition, we don’t think that Figure 1 is suitable for the proposed topic. Overall, it needs to address aforementioned issues carefully.

Answer: Figure 1 is a schematic illustration of the measuring setup which is most frequently used for measurements of the solid or liquid food samples. We think that as such it is suitable for the topic. Now we improved the Figure 1 by adding a part presenting the final results of such measurement – the obtained spectral data. In this form it is more suitable for the subject of the MS.

Comments on the Quality of English Language: Moderate editing of English language required

Answer: We additionally edited English language.

Reviewer 2 Report

The manuscript is a review paper presenting fluorescence techniques in fòod analysis. It enumerates many examples, but lacks technical detail. In each type of food some examples must be illustrated by figures and actual data in order to give a thorough description of the state-of-the art in the field.

Author Response

Reviewer 2

The manuscript is a review paper presenting fluorescence techniques in fòod analysis. It enumerates many examples, but lacks technical detail. In each type of food some examples must be illustrated by figures and actual data in order to give a thorough description of the state-of-the art in the field.

Answer: In the revised version, we included more technical details and additional Figures (Figs 3 and 4). The new text is added as track changes.

Reviewer 3 Report

The manuscript of “Intrinsic fluorescence markers for food characteristics, shelf life and safety estimation: Advanced analytical approach” summarized the fluorescence technique applied in food tests, it is good idea to review the research advances using such a simple approach to check food quality and safety. However, this review is still lack of some important highlights to attract readers. Here are some suggestions to help authors improve the paper’s quality.

(1) Figure 1 should be improved to cover the main content of this paper, for example adding the EEM vs a single excitation-emission fluorescence, etc.

(2) In the section 3, I suggest adding some figures for each sub-section to help readers understand the face-front fluorescence tests. In addition, it is better to compare and find the difference of the fluorescence data or spectra among different food types.

(3) Subtitle numbers in section 3 should be renumbered, e.g. 3.3 Honey should be 3.4 Honey.

(4) As a review paper, the number of the present references is not enough (normally >100), particularly some papers published in recent 3 or 5 years should be checked and cited regarding to the face-front or surface fluorescence spectroscopy.

English is good, some minor errors should be checked again.

Author Response

Reviewer 3

The manuscript of “Intrinsic fluorescence markers for food characteristics, shelf life and safety estimation: Advanced analytical approach” summarized the fluorescence technique applied in food tests, it is good idea to review the research advances using such a simple approach to check food quality and safety. However, this review is still lack of some important highlights to attract readers. Here are some suggestions to help authors improve the paper’s quality.

  • Figure 1 should be improved to cover the main content of this paper, for example adding the EEM vs a single excitation-emission fluorescence, etc.

Answer: In the revised version, we improved the Figure 1 by adding a part presenting the final results of such measurement – the obtained spectral data (EEM) and their processing. In this form it is more suitable for the subject of the MS.

(2) In the section 3, I suggest adding some figures for each sub-section to help readers understand the face-front fluorescence tests. In addition, it is better to compare and find the difference of the fluorescence data or spectra among different food types.

Answer: We added the new Figures in the sub-sections 3.2. and 3.3, along with the corresponding explanations.

 (3) Subtitle numbers in section 3 should be renumbered, e.g. 3.3 Honey should be 3.4 Honey.

 Answer: Corrected.

(4) As a review paper, the number of the present references is not enough (normally >100), particularly some papers published in recent 3 or 5 years should be checked and cited regarding to the face-front or surface fluorescence spectroscopy.

Answer: Thank you for this suggestion. We added more references and especially the papers published recently. The new text is added as track changes.

Comments on the Quality of English Language

English is good, some minor errors should be checked again.

Answer: We did additional check and corrections of the English language.

Reviewer 4 Report

this article summaries the different bibliographical references reporting analysis through fluorescence spectroscopy. this is a simple and fast method without sample pretreatment. this technique was developed for food samples by applying the front measurement setup.

fluorescence spectroscopy has shown great potential for studying changes in food properties during storage and detection of adulterants.

this article shows a few specific data giving an overview of food in liquid and solid state.

data should be presented in schematic tables, reporting specific references to the literature papers and to the fluorophores used for the analysis.

furthermore, the authors should therefore review the descriptive part of the samples, highlighting more which fluorophores have been used to monitor the effects of environmental factors on food structural changes.

Author Response

Reviewer 4

This article summaries the different bibliographical references reporting analysis through fluorescence spectroscopy. This is a simple and fast method without sample pretreatment. This technique was developed for food samples by applying the front measurement setup.

Fluorescence spectroscopy has shown great potential for studying changes in food properties during storage and detection of adulterants.

This article shows a few specific data giving an overview of food in liquid and solid state.

Data should be presented in schematic tables, reporting specific references to the literature papers and to the fluorophores used for the analysis.

Answer: We provided the Table showing the fluorophores used as intrinsic probes in the food samples. In the revised version, we added in the Table 1 the references to the literature papers related to the particular fluorophores shown in the Table.

Furthermore, the authors should therefore review the descriptive part of the samples, highlighting more which fluorophores have been used to monitor the effects of environmental factors on food structural changes.

Answer: In the revised version, we included more extensive explanations of the specific fluorophores which are used to monitor the effects of various factors on food structural changes. We also included additional references/examples from the recently published literature. The new text is added as track changes.

Reviewer 5 Report

The manuscript summarises most of the intrinsic fluorescence markers present in food and reviews their analytical approaches for food sample analysis. Overall this is a well-written manuscript, however, it needs minor revisions. 

1. Please correct the typo error in line 18. "excitatation"

2. Table 1, please include the respective references (citations) of the intrinsic fluorophores within Table 1 itself. 

3. In general, excitation and emission maxima depend on the solvent of the medium. It would be better if a new column about the solvent medium can be included in Table 1. 

4.  Author used very less figures in the review. They could adapt the figures from the literature if copyright is not an issue. One illustrative figure about the advanced analytical approaches should have been there. This will make it very easy to follow by all the readers without losing interest. 

Author Response

Reviewer 5

The manuscript summarises most of the intrinsic fluorescence markers present in food and reviews their analytical approaches for food sample analysis. Overall this is a well-written manuscript, however, it needs minor revisions. 

  1. Please correct the typo error in line 18. "excitatation"

Answer: Corrected.

  1. Table 1, please include the respective references (citations) of the intrinsic fluorophores within Table 1 itself. 

Answer: The respective references (citations) of the intrinsic fluorophores are now included in Table 1.

  1. In general, excitation and emission maxima depend on the solvent of the medium. It would be better if a new column about the solvent medium can be included in Table 1. 

Answer: The food samples are mostly studied without dissolution, i.e. in their natural state. It is an advantage of the optical method- it can be applied without sample pre-processing. So the solvents are not included in the Table 1.

  1. Author used very less figures in the review. They could adapt the figures from the literature if copyright is not an issue. One illustrative figure about the advanced analytical approaches should have been there. This will make it very easy to follow by all the readers without losing interest. 

Answer: We added the new Figures (Figs 3 and 4) in the section 3, to better illustrate the examples of the various food samples measured using the fluorescence spectroscopy approach. We also improved Figure 1 by adding a part presenting the final results of such measurement – the obtained spectral data. In this form it presents the potential of this analytical approach.

Round 2

Reviewer 2 Report

The manuscript has improved, the authors added some details and figures. It may be published in the present form.

Minor issue: the last phrase of the manuscript may not be complete.

Author Response

Reviewer 2

The manuscript has improved, the authors added some details and figures. It may be published in the present form.

Minor issue: the last phrase of the manuscript may not be complete.

Answer: Corrected.

Reviewer 3 Report

Part of the suggestions were not checked carefully

(1) subtitle numbers

(2) figures in section 3 are less, lack of attraction for readers

(3) In addition, it is better to compare and find the difference of the fluorescence data or spectra among different food types.

Author Response

Reviewer 3

Part of the suggestions were not checked carefully

(1) subtitle numbers

Answer: Corrected.

(2) figures in section 3 are less, lack of attraction for readers

Answer: We increased the number of figures in section 3. As this is a review paper, its value is in providing information to the readers on the state of the art in the subject field. Some examples are given in the Figures.

(3) In addition, it is better to compare and find the difference of the fluorescence data or spectra among different food types.

Answer: Thank you for the comment. We agree that there are different ways to organize the text, i.e. how to present the examples. We chose to present the applications of the method for particular food types. Since different food types differ in physical and chemical characteristics, it would not be quite appropriate to compare the spectra among them.
